# Spatial Segregation Within Dissolving Microneedle Patches Overcomes Antigenic Interference and Enables Potent Bivalent Influenza–RSV Vaccination in Mice

**DOI:** 10.3390/vaccines13121213

**Published:** 2025-11-30

**Authors:** Feng Fan, Yehong Wu, Hongzhe Lin, Xin Zhang, Limei Wang, Yue He, Shijie Zhang, Mingju Zhang, Gan Zhao, Rong Xiang, Yating Kang, Mingyue Chen, Zhuang Li, Yi-Bing Guo, Hang Zhou, Chen Zhao, Man-Chuan Wang, Jian-Yang Gu, Bin Wang, Xiao-Ming Gao

**Affiliations:** 1Advaccine Biopharmaceutics (Suzhou) Co., Ltd., Suzhou 215123, China; 2Changchun Institute of Biological Products Co., Ltd., Changchun 130012, China; 3Key Laboratory of Medical Molecular Virology (MOE/NHC/CAMS), Shanghai Institute of Infectious Disease and Biosecurity, School of Basic Medical Sciences, Fudan University, Shanghai 200032, Chinabwang@advaccine.com (B.W.); 4School of Life Science and Technology, Changchun University of Science and Technology, Changchun 130022, China

**Keywords:** intra-vaccine interference, influenza, RSV, microneedle patch, partition loading

## Abstract

**Background/Objectives:** Given the overlapping seasonality of influenza (Flu) and respiratory syncytial virus (RSV) infections in human populations, Flu–RSV combination vaccines are urgently needed. However, development of combo-vaccines is often faced with intra-vaccine interference which could compromise vaccination outcomes. Here we present an approach to overcoming this problem using a microneedle array patch (MAP)-based combo-vaccine with minimum intra-vaccine interference. **Methods:** Vaccine-laden dissolving MAPs were fabricated using a two-step micro-molding process with polyvinyl alcohol as major excipient. A partition-loading strategy was adopted to ensure spatially segregated distribution of a split-virus Flu vaccine and recombinant prefusion protein of RSV in separate MAP sectors. Serum samples from BALB/c mice post-vaccination were assessed for titers of binding and neutralizing antibodies against the viruses. Live virus challenge studies were carried out to assess the protection efficacy of the MAP-based vaccines. **Results:** Although i.m. administered standalone Flu and RSV vaccines were able to induce strong IgG responses in BALB/c mice, bidirectional intra-vaccine interference was observed when the two vaccines were co-administered in premixed form. However, when the two vaccines were loaded onto nonoverlapping sectors of D-MAPs for intradermal vaccination, the intra-vaccine interference effect was effectively overcome. The partition-loaded MAP-Flu/RSV combo-vaccine elicited antigen-specific IgG with robust virus-neutralizing activity and was strongly efficacious against either virus in challenge studies. **Conclusions:** Our data provide proof-of-concept evidence for the potential usefulness of partition-loaded MAPs in overcoming a critical barrier in vaccinology and offer a promising platform for future clinical translation.

## 1. Introduction

Influenza (Flu) virus and respiratory syncytial virus (RSV) are both single-stranded negative-sense RNA viruses with genomes each encoding 10 to 11 canonical proteins, though additional products may arise from alternative splicing or overlapping reading frames [1,2]. They may cause respiratory tract illness in all ages, but the young and the elderly are particularly vulnerable [3]. Influenza causes an estimated 1 billion infections and 3–5 million severe cases annually, while RSV causes >33 million lower-respiratory infections in young children each year, resulting in an immense public health burden [1,2,3]. Three types of licensed seasonal Flu vaccines have been approved for human use: inactivated, live attenuated, and recombinant hemagglutinin (HA) vaccines. Nowadays the inactivated Flu vaccine is mostly prepared in the form of a split virus containing mainly HA [4]. The licensed trivalent Flu vaccine from Changchun Institute of Biotechnological Products (CIBP) Co., Changchun, China is a split-virus vaccine produced by detergent disruption of purified influenza virions of the A/Darwin (H3N2), A/Vitoria (H1N1), and B/Austria (BV) strains to remove the viral envelope. HA1 and HA2 subunits of HA constitute the dominant protective components of the resultant CIBP trivalent Flu vaccine.

So far GSK’s Arexvy, Pfizer’s Abrysvo, and Moderna’s mRESVIA vaccines have been licensed for active immunization against RSV in adults aged 60 years or more and for passive immunization of infants by maternal administration during pregnancy [5,6]. The F, or prefusion stabilized F (preF), glycoprotein of RSV is targeted by these vaccines due its conserved nature and necessity for virus entry [7,8]. Additionally, it has been documented that preF protein can enhance the anti-RSV-G protein antibody and anti-infection responses in vivo [9].

Given the overlapping seasonality of the Flu and RSV infections, their coinfection can occur and cause a more severe disease outcome [3,10]. Therefore, combination vaccines (ready-to-use multivalent vaccine units that are administered into a single site) for Flu and RSV are urgently needed. Preclinical studies on bivalent Flu–RSV combo-vaccine candidates have made much progress in recent decades. For example, a Flu–RSV dual subunit vaccine consisting of Flu HA and RSV preF5 exhibited robust immunogenicity and efficacy against Flu and RSV challenge in mice [11,12]. More recently, another recombinant bivalent Flu–RSV vaccine was found to be effective in protecting BALB/c mice from challenges by either virus [13]. Despite these advances, no Flu–RSV combo-vaccines have been licensed so far.

One of the problems facing combination vaccines is intra-vaccine interference, which is defined as the negative effect of one component within the vaccine formulation on the vaccinee’s response to the other component(s) [14,15,16,17]. This phenomenon can be evaluated by comparing the antibody (Ab) responses elicited by the combo-vaccine and the individual components alone [15,16,17]. Molecular mechanisms behind intra-vaccine interference remain poorly understood. One possible reason is the interaction or complexation between vaccine components during formulation, although host factors can also play an important role in influencing immunogenicity of combo-vaccines. While most studies on intra-vaccine interference primarily assess humoral responses, the impact on cellular immunity remains poorly defined.

Most current vaccines are delivered by i.m. or s.c. injection using needles, which need trained professionals for administration and often cause pain at the injection sites. To improve vaccine administration, various types of microneedle array patches (MAPs) have been developed for convenient intradermal vaccination without using hypodermic needles [18,19,20,21]. Dissolving MAPs (D-MAPs), fabricated using biocompatible and dissolvable high molecular weight materials as major excipients, are perhaps the most favored choice for intradermal vaccine delivery. Once inside the epidermis, microneedle tips of D-MAPs readily dissolve to release the vaccine load within minutes [18,19]. Recent preclinical and clinical studies have shown that D-MAPs can effectively deliver vaccines against various human viruses. For example, in a phase I clinical trial by Rouphael et al. a D-MAP-based Flu vaccine was shown to be safe and immunogenic in adults [22]. D-MAPs have also been shown to strongly improve the immunogenicity of inactivated poliovirus vaccine (IPV) and inactivated rotavirus vaccine (IRV) in rhesus macaques and mice [23,24]. Joyce et al. reported an immunogenic and protective D-MAP-based measles and rubella vaccine in infant rhesus macaques [25]. We recently reported robust immunogenicity of D-MAP-based DNA vaccine candidates against SARS-CoV-2 in mice, rabbits, and pigs [26,27]. Based on these promising results, the present study aimed to develop D-MAPs laden with Flu–RSV combo-vaccine. To overcome intra-vaccine interference, we designed a partition-loading strategy, which physically separated Flu and RSV vaccines within a single patch, and then assessed immunogenicity and efficacy of the vaccine. The results provide proof-of-concept evidence for the potential usefulness of partition-loaded MAPs in combo-vaccine development.

## 2. Materials and Methods

### 2.1. Flu and RSV Vaccines Used in This Study

The licensed trivalent Flu vaccine from CIBP Co., Changchun, China is a split-virus vaccine produced by detergent disruption of purified influenza virions of the A/Darwin, A/Vitoria, and B/Austria strains to remove the viral envelope. HA1 and HA2 subunits of HA constitute the dominant protective components of the vaccine.

Advaccine Biopharmaceutics Co., Ltd. (Advaccine, hereafter) licensed a preF patent from NIH, USA for conducting studies to develop the RSV vaccine. The vaccine construct encoded recombinant prefusion F protein with an ectodomain sequence (amino acid residues 1 to 513) from RSV subgroup A2 (rpF5), fused with a T4 fibritin foldon trimerization domain. For expression of the recombinant fusion protein, the DNA construct was expressed by transfection of Expi293F cells (Thermo Fisher Scientific, Waltham, MA, USA) using 293fectin (Thermo Fisher Scientific, Waltham, MA, USA). The culture supernatant, after centrifugation to remove cell debris, was sterile-filtered, from which rpF5 was purified sequentially by affinity chromatography and size-exclusion chromatography (GE Healthcare, Bensalem, PA, USA). SDS-PAGE electrophoresis confirmed that the end product was more than 95% pure rpF5 protein. The purified protein was dissolved in PBS containing 0.1 PS-80 and stored at −80 °C until use.

### 2.2. Fabrication of D-MAPs Laden with Flu, RSV, or Flu–RSV Vaccines

D-MAPs used in this study were prepared by a two-step micro-molding process, as previously described [26]. Briefly, the D-MAP-based vaccine formulation consisting of inactivated Flu split-virus and/or RSV rpF5, water-soluble high molecular weight polyvinyl alcohol (PVA), supplemental sucrose, and salts was cast onto a polydimethylsiloxane (PDMS) mold. The top layer of partition loadings was performed using a programmable and automated nanoliter dispensing system, featuring high-precision nozzles and integrated into a computerized dispensing machine. This system was designed by the Advaccine medical team in collaboration with external partners, who customized the dispensing machine with tailored parameters for the final manufacturing process. To fabricate a D-MAP-based Flu–RSV combo-vaccine using the partition-loading strategy, the Flu and RSV vaccines were individually cast onto prearranged nonoverlapping sectors of the PDMS mold. The formulation was allowed to air dry at room temperature overnight. Then the backing formulation was cast onto the mold and subsequently dried at room temperature for 4 h before demolding the MN patch, which was further mounted onto a 1.2 cm^2^ paper backing, followed by packaging in aluminum–plastic bags and stored at +4 °C until use.

### 2.3. Western Blot

The Flu split-virus vaccine and RSV rpF5 were separated by SDS-PAGE 10% gels and transferred to polyvinylidene difluoride (PVDF) membranes. Immunoblotting was performed by using sheep antisera against Flu HA (National Institute for Biological Standards and Control (NIBSC), Potters Bar, Hertfordshire, UK) diluted 1:1000 in 5% milk–0.05% PBS–Tween 20. HRP-labeled rabbit anti-goat IgG (ZhongShan Golden Bridge Biotechnology Co., Ltd., Beijing, China), was employed as secondary Ab. Chemiluminescence detection was performed with the ECL Prime Western Blotting System and acquired by the ChemiDoc Imaging System (Bio-Rad, Hercules, CA, USA).

### 2.4. Animal Immunization

Female BALB/c mice (6–8 weeks of age) were purchased from Shanghai Slac Laboratory Animal Co., Ltd. (Shanghai, China) or Beijing Vital River Laboratory Animal Technology Co., Ltd. (Beijing, China) and maintained under SPF conditions at the animal facilities of Advaccine Biologics (Suzhou, China) Co, CIBP, or Fudan University. All animal experiments were performed in compliance with the recommendations in the Guide for the Care and Use of Laboratory Animals of the Ministry of Science and Technology Ethics Committee and approved by the Ethics Committees of Advaccine, CIBP, or Fudan University. All mice were euthanized by cervical dislocation at the end of the experiments.

To administer MAP-based vaccines in mice, larger-than-patch-sized dorsal skin sites were shaved and treated with hair removal cream one day earlier. The shaved mice were placed in the induction chamber of an RWD small animal anesthesia machine, and anesthetized with 5% isoflurane (Ringpu Biology, Tianjin, China) for 2–3 min. The fully anesthetized mice were then ready for microneedle patch administration. The patch was then applied and gently pressed with thumb pressure on the shaved skin surface of the animals under anesthesia. It was allowed to stay for 15 min and the used patch was then removed.

### 2.5. Enzyme-Linked Immunosorbent Assay

Antibody titration was performed on sera obtained by retro-orbital bleeding from the mice. ELISA plates were coated with Flu split-virus or RSV rpF5 protein at 1 mg/mL and incubated 18 h at 4 °C and subsequently blocked with 3% BSA–0.05% Tween 20–PBS (PBST) for 1 h at room temperature. Serially diluted serum samples were then added in triplicate wells, and the plates were incubated for 1 h at room temperature. After a double wash with PBST, horse-radish peroxidase (HRP)-conjugated Ab against murine IgG (Abcam, ab6789, Cambridge, UK) was added and then developed with 3,3′,5,5′-tetramethylbenzidine (TMB) substrate (Coolaber, Beijing, China). The reaction was stopped with 2 M of H_2_SO_4_, and the absorbance measured at 450 nm and reference 620 nm using a microplate reader (TECAN, Männedorf, Switzerland). Serum endpoint titers were defined as the dilution where OD values dropped to <2x the normal mouse serum control.

### 2.6. RSV Neutralization Assay

RSV neutralization titers were determined using a microtiter-based assay system. Serially diluted antisera samples were mixed with a standardized RSV inoculum, dispensed into 96-well microtiter plates containing adherent HEp-2 cells, and then incubated for 2–3 days at 37 °C in a 5% CO_2_ humidified incubator. Positive control wells (RSV-infected, serum-free) were included to monitor maximal viral infectivity. Cytopathic effect (CPE) development was monitored until clear viral CPE became evident in control wells. Plates were then washed three times with PBST, fixed with cold 80% acetone in PBS, and blocked with 3% (*w*/*v*) blocking buffer. Detection of virus replication/inhibition was performed using goat anti-RSV polyclonal antibody (Meridian Life Science, Saco, ME, USA) followed by HRP-conjugated bovine anti-goat IgG secondary antibody (Santa Cruz Biotechnology, Santa Cruz, CA, USA). After chromogenic substrate development, absorbance was measured at dual wavelengths (450/620 nm) using an ELISA plate reader. Neutralizing antibody titers were calculated as the reciprocal of the highest serum dilution that reduced RSV-induced CPE by ≥50% compared to positive controls, with endpoint titers determined by interpolation from the dose–response curve based on mean OD values of triplicate wells.

### 2.7. Hemagglutination Inhibition Assay

Hemagglutination inhibition (HI) assays were performed to determine antibody titers against Flu virus A/Victoria (H1N1), A/Darwin (H3N2), and B/Austria (B/Victoria lineage) strains. Serum samples were treated with 80 μL of cholera filtrate (Sigma-Aldrich, St. Louis, MO, USA) at 37 °C for 18 h, heat-inactivated at 56 °C for 50 min, and adsorbed with 10 μL packed chicken red blood cells at 4 °C overnight. Influenza antigens (Changchun Institute of Biological Products Co., Ltd., Changchun, China) were standardized to 4 hemagglutinating units. Two-fold serial dilutions of sera were mixed with antigen and incubated for 45 min at room temperature, followed by addition of 1% chicken RBCs. HI titers were read after 45 min as the highest serum dilution that completely inhibited hemagglutination.

### 2.8. Live RSV Challenge In Vivo and RSV Load Determination by PCR

For the live RSV challenge study (Project ID: BSL2A-2024-0027), female BALB/c mice aged 6–8 weeks, purchased from Beijing Vital River Laboratory Animal Technology Co., Ltd., were housed in Fudan University’s BSL-2+ laboratory. Two weeks after immunization, the mice were anesthetized with i.p. injection of 300 μL (20 mg/mL) tribromoethanol (Nanjing Aibei Biotechnology Co., Ltd., Nanjing, China) and then intranasally (i.n.) inoculated with 50 μL of RSV A (long strain) virus at a concentration of 3.16 × 10^6^ TCID50/mL. The infected mice were monitored daily for bodyweight changes until 5 days post-infection (DPI), and then sacrificed by cervical dislocation. We harvested the lung. Left lobes were used for histopathology and the remnants were used for viral load determination by RT-qPCR. Total cellular RNA, extracted from the lung tissue homogenates using a viral RNA extraction kit, was reverse-transcribed into cDNA which was employed as template for qPCR detection of the RSV N gene using forward and reverse primers 5′-ttacggtggggagtcttagc-3′ and 5′-cactggagaagtgaggaaattgag-3′, respectively. For absolute quantification, a standard curve was generated using 10-fold serial dilutions (10^0^–10^6^ copies/reaction) of a plasmid DNA construct containing the RSV N gene. Viral loads in samples were calculated directly from the standard curve and reported as RSV RNA copies per mg RNA extracted from the RSV-challenged animal. Reaction conditions included initial denaturation at 95 °C for 30 s, followed by 40 cycles of 95 °C for 10 s and 60 °C for 30 s.

### 2.9. Live Flu Virus Challenge In Vivo

Two weeks after the final immunization, BALB/c mice were anesthetized with a tiletamine–zolazepam cocktail (Zoletil 50, 5 + 5 mg/mL, 100 µL/mouse i.p.) and then i.n. challenged with 100 μL of live influenza virus containing 10 × LD_50_ of either A/Victoria/4897/2022 or B/Austria/1359417/2021, both maintained at CIBP Co., Ltd. (Changchun, China). Mice were observed daily for 14 consecutive days post-infection (DPI) for bodyweight changes and survival. Animals that lost more than 25% of their initial body weight or exhibited severe clinical signs were humanely euthanized in accordance with institutional animal care and use guidelines.

### 2.10. Statistics

Statistical analyses were performed with GraphPad Prism software version 9 (GraphPad). Error bars indicate the standard error of the mean (SEM). We used Mann–Whitney *t*-tests to compare two groups with non-normally distributed continuous variables and two-way ANOVA followed by Sidak’s multiple comparisons tests to analyze experiments with multiple groups and two independent variables. Significance is indicated as follows: * *p* < 0.05; ** *p* < 0.01. Comparisons are not statistically significant unless indicated.

## 3. Results

### 3.1. Antigenic Characteristics of the Trivalent Inactivated Flu and the Recombinant PreF RSV Vaccines

Samples of the CIBP trivalent Flu vaccine were run on SDS-PAGE gels followed by Western blotting using Abs specific for HA1/HA2 of the Flu virus A/Victoria (H1N1), or A/Darwin (H3N2), or B/Austria (BV) strains. As expected, protein bands of approximately 55–60 kDa and 25 kDa, equivalent to the molecular masses of HA1 and HA2, respectively, were strongly recognized (Figure 1A). The recombinant RSV preF5 protein (rpF5) was prepared and purified by Advaccine as a candidate RSV vaccine. Several stabilization strategies were incorporated in the design of rpF5, including (i) two strategically placed disulfide bonds (Cys155–Cys290 and Cys149–Cys458) to reinforce key structural domains; (ii) proline substitutions at position 215 to restrict conformational flexibility; (iii) cavity-filling mutations (S190F and V207L) to optimize antigenic site stability; and (iv) a C-terminal fibritin trimerization domain to ensure native-like trimer formation under physiological conditions. SDS-PAGE gel electrophoresis revealed protein bands of approximately 65 kDa and 195 kDa in the rpF5 preparation under reducing condition and non-reducing conditions, respectively (Figure 1B). Apparently, trimerization occurred to the rpF5 protein under non-reducing conditions, which is a known characteristic of the preF protein of RSV (8, 11, 12).

### 3.2. Immunogenicity of the Flu and rpF5 Vaccine Preparations in Mice

To assess the immunogenicity of the Flu and RSV vaccines, groups of BALB/c mice were i.m. immunized twice with either Flu A/Darwin split-virus (0.3 μg or 1.5 μg per dose), or rpF5 (10 μg/dose). The Flu vaccine was administered without adjuvant, while the rpF5 recombinant protein was given in alum-adjuvated, or adjuvant-free, form. Serum samples collected on day 28 were analyzed by ELISA to determine IgG binding titers against the immunizing antigens. While unimmunized control mice did not generate detectable antigen-binding Abs, Flu vaccine-immunized mice exhibited high antigen-specific IgG titers (1:178,200 ± 132,800 and 1:469,800 ± 182,133), while rpF5/alum-immunized mice showed even stronger responses (1:5,511,240 ± 1,574,640) (Figure 2). Additionally, the HI titer of 1.5 μg/dose Flu antisera exceeded 1:512, and the RSV-neutralizing titer of the rpF5/alum-immunized group reached 1:256 (Figure 2). These findings reveal distinct immunogenic characteristics between the two vaccines. Notably, the unadjuvanted Flu vaccine induced comparably high binding-IgG titers at 0.3 μg and 1.5 μg doses (Figure 2A), though the higher dose appeared more effective for generating HI Abs. In contrast, the rpF5 protein required alum adjuvant to achieve significant antibody responses at the tested 10 μg dose (Figure 2B). Based on these differential dose–adjuvant requirements, we selected 1.5 μg Flu vaccine and 10 μg rpF5 protein for subsequent combination vaccine formulation and immunization studies.

### 3.3. Intra-Vaccine Interference of the Premixed Flu–RSV Combo-Vaccine

We next formulated an aluminum-adjuvanted combination vaccine by mixing the Flu A/Darwin split-virus and the RSV rpF5 protein at a 1.5:10 ratio. The resultant H3N2–rpF5 combo-vaccine was compared with the standalone A/Darwin split virus, or rpF5, for ability to induce antigen-specific IgG in BALB/c mice following two doses of i.m. administration. Interestingly, titers of the antigen-specific serum IgG generated by the combo-vaccine were significantly lower than that by either the Flu H3N2 or RSV rpF5 immunogen alone (Figure 3). Titration curves for individual serum samples of these immunization groups are shown in Appendix A. These results indicate bidirectional interference between the Flu and RSV vaccines when i.m. administered in premixed form. A possible reason for this phenomenon would be that, once mixed together, the two vaccine components complexed with each other, which lead to insufficient antigen up-taking and processing in vivo. However, HPLC analysis of the premixed H3N2 and rpF5 vaccines did not find evidence for direct complexation between them (Appendix A). More in-depth biochemical and immunological investigations would be required before a definitive conclusion can be drawn.

### 3.4. Partition-Loading Strategy for D-MAP-Based Combo-Vaccine Fabrication

Previous investigators have documented fabrication of various D-MAP-based combo-vaccines consisting of either inactivated viruses or recombinant proteins that exhibited good immunogenicity in preclinical and clinical studies [24,25]. However, those combo-vaccines were loaded onto MAPs in premixed form, which may still face the problem of intra-vaccine interference. Here we designed a partition-loading strategy for fabrication of D-MAP-based combination vaccines, which can ensure nonoverlapping distribution of different vaccine components in separate patch sectors (Figure 4A). To demonstrate the technique feasibility of this strategy, various combinations of colorants (grey, red, blue, or green) as vaccine mimics were partition-loaded onto single 1 cm^2^ D-MAPs in predesigned patterns (Figure 4B–D). The mechanical integrity and penetration effectiveness of the fabricated patches were confirmed by parafilm pack penetration tests (Appendix A). When the colorant-laden D-MAPs were applied on the dorsal skin of BALB/c mice for 10–15 min followed by removal of the used patches, sharp non-overlapping intradermal color imprints were left on the treated skin sites (Figure 4E–G). These results suggest that different vaccine components can be accurately uploaded onto nonoverlapping sectors of D-MAPs, which should be able to deliver components of combo-vaccines into non-overlapping areas of the patch-applied skin sites.

### 3.5. Immunogenicity Assessment of the MAP-Based Flu–RSV Combo-Vaccine In Vivo

Next, by using the partition-loading strategy, we fabricated D-MAPs carrying 1.5 μg Flu A/Darwin split-virus and 10 μg RSV rpF5 protein (MAP-H3N2/rpF5). Meanwhile, D-MAPs carrying the Flu (H3N2) and RSV (rpF5) antigens, either premixed or standalone, were also prepared (Figure 5A). The resultant MAP-based vaccines were then compared with i.m. delivered A/Darwin split-virus and rpF5 antigen for ability to elicit antigen-specific IgG responses in BALB/c mice. In agreement with the data shown in Figure 3, i.m. delivered H3N2/rpF5 mixture demonstrated significantly inferior immunogenicity compared to either antigen administered alone (Figure 5B,C). Importantly, partition-loaded MAP co-delivery completely abolished this intra-vaccine interference. The MAP-H3N2/rpF5 formulation induced antigen-specific serum IgG titers that were comparable to—or even higher than—those achieved by monovalent H3N2 and rpF5 vaccines delivered either intramuscularly or via MAP. Notably, while the MAP-H3N2/rpF5mix formulation also significantly outperformed conventional i.m. co-administration of the H3N2/rpF5 mixture, its IgG response magnitude remained lower than that of the partition-loaded MAP-H3N2/rpF5 group (though the difference did not reach statistical significance). The following conclusions can be drawn from these results: (i) D-MAPs carrying killed Flu virus and rpF5 protein exhibit robust immunogenicity in vivo, and (ii) the partition-loaded patches circumvent the intra-vaccine interference observed with premixed formulations.

### 3.6. Protection Efficacy of the MAP-Flu/RSV Combo-Vaccine Against Live Virus Challenge in Mice

The above results prompted us to fabricate another D-MAP combo-vaccine (MAP-Flu/RSV) carrying the CIBP trivalent Flu vaccine (comprising purified split-virus of the Flu virus A/Darwin, A/Vitoria, and B/Austria strains in premixed form) and RSV rpF5 recombinant protein for protection efficacy testing. The trivalent Flu and the rpF5 RSV antigens were partition-loaded onto D-MAPs at a 1.5 μg:10 μg ratio, with each vaccine component occupying one non-overlapping half of the patch. Additionally, D-MAPs laden with a 1.5 μg:10 μg mixture of the trivalent Flu and the rpF5 antigens (MAP-Flu/RSVmix), or standalone 1.5 μg trivalent Flu vaccine (MAP-Flu) were also fabricated as additional control samples. No adjuvant was included in these vaccine patches.

As shown in Figure 6A, RSV-neutralizing Ab titer in BALB/c mice after MAP-Flu/RSV immunization was comparable to that induced by MAP-rpF5 or IM-rpF5 (day 35), yet higher than that of the MAP-Flu/RSVmix group (1/512 vs. 1/126). Following live RSV i.n. challenge, both vaccinated and unvaccinated mice exhibited only mild, transient weight loss following infection (Figure 6B). However, by 5 DPI, MAP-Flu or IM-PBS control mice showed substantial pulmonary viral loads (3500–4100 viral copies per mg RNA), in stark contrast to the MAP-Flu/RSV, MAP-rpF5, and IM-rpF5 groups, which achieved complete viral clearance (Figure 6C). While MAP-Flu/RSV-immunized mice showed significant viral load reduction compared to controls, their lung viral clearance remained incomplete. The marked protection disparity between the MAP-Flu/RSV and MAP-Flu/RSV vaccines provides further evidence for immunological interference between the Flu HA and RSV preF antigens when administered in premixed form.

The partition-loaded MAP-Flu/RSV patches effectively induced cross-neutralizing serum Abs in BALB/c mice against multiple Flu strains, including A/Victoria, A/Darwin, and B/Austria strains. As shown in Figure 7A–C, the HI titers achieved by MAP-Flu/RSV immunization were similar to those induced by either MAP-Flu or IM-Flu vaccination. Following lethal i.n. challenge with either A/Victoria or B/Austria Flu viruses, control mice receiving MAP-placebo or IM-PBS suffered approximately 20% bodyweight loss and 80% mortality 5–7 DPI (Figure 7D–G). By contrast, mice given MAP-Flu/RSV, MAP-Flu, or IM-Flu vaccination showed only mild transient weight loss and achieved 80–100% survival rates by 12 DPI (Figure 7D–G). Together, these data support the idea that the partition-loaded MAP-Flu/RSV vaccine not only overcomes potential intra-vaccine interference but also delivers robust protective efficacy against both influenza virus and RSV in vivo.

## 4. Discussion

Employing our partition-loading technique, we developed D-MAP-based Flu/RSV combination vaccines (MAP-H3N2/rpF5 and MAP-Flu/RSV) that elicited robust protective immunity against both pathogens in vivo. Notably, these partition-loaded formulations demonstrated comparable or superior immunogenicity and/or protective efficacy compared to individually administered or premixed Flu and RSV vaccines in BALB/c mouse models (Figure 5, Figure 6 and Figure 7). These findings provide compelling proof-of-concept for partition-loaded D-MAPs as an effective strategy to overcome intra-vaccine interference—a fundamental challenge in combination vaccine development. The demonstrated efficacy suggests strong potential for MAP-Flu/RSV vaccines in human immunization programs, particularly during winter seasons when concurrent influenza and RSV circulation poses significant public health risks.

Partition-loaded MAPs offer three key advantages for combination vaccine development. First, they provide exceptional formulation flexibility by enabling compartmentalized loading of diverse vaccine types within a single patch. Unlike conventional formulations that preclude combining different vaccine modalities (e.g., live-attenuated/killed viruses, recombinant proteins, toxoids, polysaccharides, and nucleic acids) in a single dose, our partition-loaded D-MAP technology successfully integrates these components. This versatility is demonstrated not only in our MAP-Flu/RSV combo-vaccine but also in D-MAPs carrying split Flu virus, rpF5, and pAD1016 (a COVID-19 DNA vaccine) [26,27], with the resultant MAP-Flu/RSV/CoV2 patch inducing robust IgG responses against all three pathogens in BALB/c mice (Appendix A). Second, partition-loaded D-MAPs simplify quality control and quantitative analysis. The patch’s compartmentalized design allows for precise segmentation and independent testing of individual vaccine components, facilitating more reliable quality assessment of complex combination formulations. Third, partition-loaded MAP patches confer robust immunogenicity without requiring traditional adjuvants, effectively addressing the critical challenge of adjuvant selection for multi-component combination vaccines. This advantage is particularly evident when considering vaccines with divergent adjuvant requirements—such as the influenza vaccine (which typically does not require adjuvants) versus the poorly immunogenic rpF5 protein that necessitates alum adjuvant for i.m. delivery (Figure 2B and Figure 5B). Conventional combination approaches would face significant difficulties in identifying a universally suitable adjuvant. Remarkably, our unadjuvanted MAP-Flu/RSV and MAP-H3N1/preF5 patches demonstrate rpF5 immunogenicity comparable to alum-adjuvanted i.m. injection (Figure 2, Figure 5, Figure 6 and Figure 7 and Appendix A). This enhanced immunogenicity likely arises from the intrinsic “adjuvant effect” of microneedle-mediated skin penetration. The physical stimulation from microneedle insertion activates dermal tissue cells to secrete pro-inflammatory cytokines, which in turn augment antigen uptake and immune activation. This mechanism is supported by our finding that placebo MAP application significantly upregulated IL-1β, IL-6, and TNF-α gene expression in mouse skin within just one hour (Appendix A).

Like Flu virus HA, the RSV F protein is a major viral glycoprotein targeted by neutralizing Abs [7]. PreF5 is also known as a molecular adjuvant able to trigger Toll-like receptor 4 (TLR4) and CD14 signaling for initiation of innate immune response at the site of delivery [8,9]. It was therefore anticipated that rpF5 of RSV might augment immune responses to the Flu–RSV combo-vaccine. Surprisingly, however, intra-vaccine interference (at least in terms of suppressed humoral responses), rather than synergy, occurred when the premixed Flu virus and rpF5 combination was i.m. administered in BALB/c mice (Figure 3, Figure 5 and Figure 6). A possible explanation for the compromised immunogenicity of the premixed Flu–RSV vaccine preparation is direct interaction between the Flu and RSV proteins, albeit further biochemical and biological investigations will be needed for a definitive conclusion.

Despite these compelling results, a few limitations remain in the present study: (i) Immunization and protection experiments were conducted exclusively in 10–12-week-old female BALB/c mice, potentially limiting the translatability of findings to pediatric or geriatric populations. (ii) Since booster vaccination significantly enhanced influenza virus-neutralizing antibody responses in our BALB/c mouse model (Appendix A), all protection data were derived from a two-dose regimen (days 0 and 21). Given that real-world adult vaccination programs typically employ single-dose administration, evaluation of a single-dose partition-loaded MAP regimen would provide more clinically relevant data. (iii) Protection experiments did not include RSV-B or Flu A/Darwin strains due to material availability constraints. Comprehensive evaluation against additional viral strains would strengthen the evidence for broad-spectrum protective efficacy. (iv) The potential enhancement of T cell responses following partition-loaded MAP-Flu/RSV versus premixed MAP-Flu/RSV-mix vaccination has not been investigated. Given that T cell-mediated immunity plays a critical role in antiviral protection, it is plausible that partition-loaded MAP vaccines mitigate intra-vaccine interference—at least in part—through modulation of cellular immune responses, though this hypothesis requires experimental validation.

Refinement of the vaccine partition-loading technique for D-MAP-based combo-vaccine fabrication requires further optimization of the microneedle array settings and automated dispensing of the vaccine loads. The use of an automated nano-dispenser allows for highly accurate and controlled deposition of vaccine components within each patch sector, minimizing the risk of cross-interaction between vaccines. The process can be adapted for high-throughput production by utilizing multi-cavity molds and parallel dispensing systems with multiple nozzles.

## 5. Conclusions

The partition-loading technique for D-MAP-based combination vaccine fabrication enables compartmentalized encapsulation of multiple vaccine types within a single patch. Our MAP-Flu/RSV combo-vaccine demonstrated superior immunogenicity and protective efficacy compared to conventional i.m. administered premixed Flu + RSV vaccine in murine models. These findings provide compelling proof-of-concept for partition-loaded MAPs as an innovative solution to a longstanding vaccinology challenge, while establishing a robust platform for clinical translation.

## Figures and Tables

**Figure 1 vaccines-13-01213-f001:**
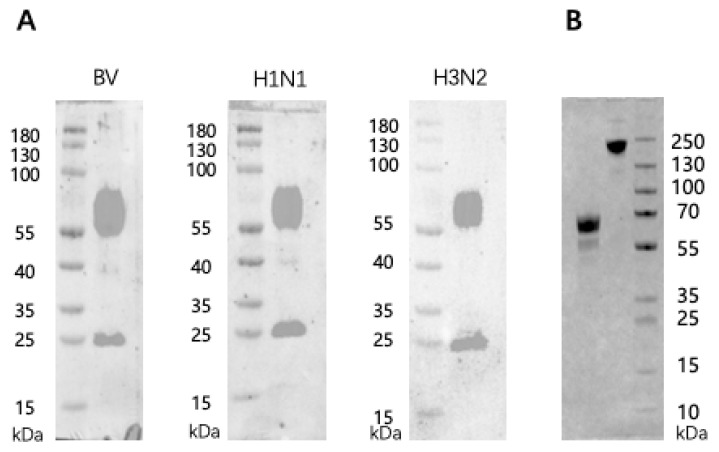
Detection of specific antigens in the Flu and RSV vaccine preparations. (**A**) Samples of the CIBP trivalent Flu split-virus vaccine were run on SDS-PAGE 10% gels under reducing conditions followed by Western blotting using sheep antisera specific for Flu HA of B/Austria (BV), A/Victoria (H1N1), or A/Darwin (H3N2) strains. HRP-labeled rabbit-anti-goat IgG was employed as secondary detection Abs. (**B**) Samples of RSV rpF5 were run on SDS-PAGE 10% gels under reducing (lane on left) or non-reducing (lane on right) conditions followed by Coomassie blue staining. Quantitative analysis using Image J revealed the following band/background intensity ratios: for the Flu HA1 bands (55–60 kDa), ratios ranged from 2.64 to 3.13; for the HA2 bands (25 kDa), ratios ranged from 1.75 to 2.64. The 195 and 69 kDa rpF5 bands exhibited band/background intensity ratios of 1.70 and 1.92, respectively.

**Figure 2 vaccines-13-01213-f002:**
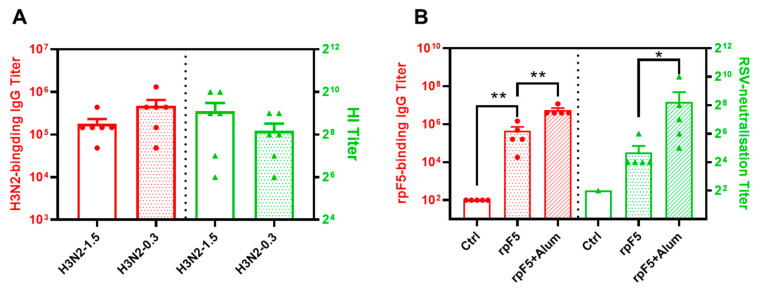
Immunogenicity of the Flu and RSV vaccines in mice. Groups of BALB/c mice were i.m. immunized twice with either the Flu A/Darwin split-virus (0.3 μg or 1.5 μg per dose, *n* = 6) (**A**), or rpF5 of RSV (10 μg/dose with or without alum adjuvant, *n* = 5) (**B**). Mice i.m. injected with PBS/adjuvant were included as adjuvant control (Ctrl). Serum samples, collected at 14 days post-immunization, were individually titrated for titers of binding IgG against the immunizing antigens in ELISAs, and further assayed for HI (**A**), or RSV neutralization (**B**), activities. The data are group means ± SEM, *p* values were analyzed with two-tailed Mann–Whitney test (*: *p* < 0.05; **: *p* < 0.01).

**Figure 3 vaccines-13-01213-f003:**
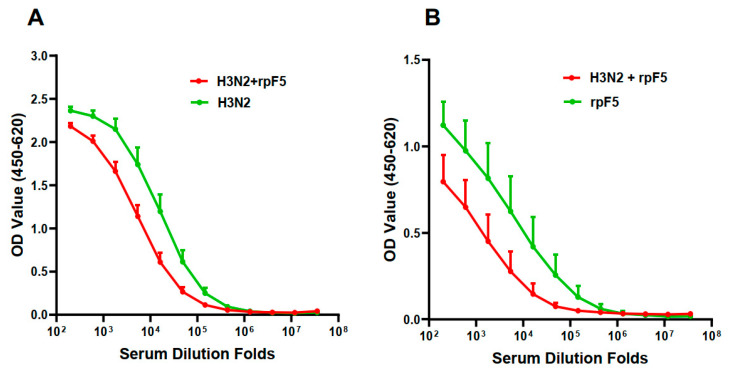
Intra-vaccine interference of a premixed H3N2/rpF5 preparation. Groups of BALB/c mice (*n* = 5) were given two doses of i.m. immunization with either the premixed Flu A/Darwin split-virus and rpF5 (1.5 μg + 10 μg/dose) combo-vaccine (H3N2/rpF5), or standalone Flu A/Darwin split-virus (1.5 μg/dose), or rpF5 (10 μg/dose, alum-adjuvanted) on days 0 and 21. Serum samples, collected from the immunized animals 14 days later, were individually titrated against the immunizing Flu (**A**) or RSV (**B**) antigen in ELISAs. The results are group means ± SEM absorbance at 450–620 nm.

**Figure 4 vaccines-13-01213-f004:**
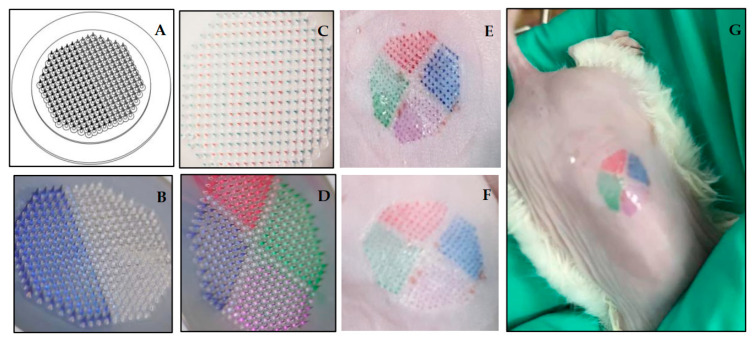
Design and feasibility of API partition-loading in D-MAP fabrication. (**A**) Illustration of a partition-loading design for D-MAP carrying dual vaccine components distributed in non-overlapping halves of a patch. (**B**–**D**) Sample D-MAPs laden with 2 or 4 different colorants as combo-vaccine mimics distributed in nonoverlapping patch sectors. (**E**–**G**) Photographs showing the skin site of a BALB/c mouse after D-MAP delivery (the used patch was removed after 15 min application) of multiple colorants in clearly non-overlapping fashion. ((**E**): Immediately after patch removal without cleaning; (**F**,**G**): after cotton swab cleaning following patch removal).

**Figure 5 vaccines-13-01213-f005:**
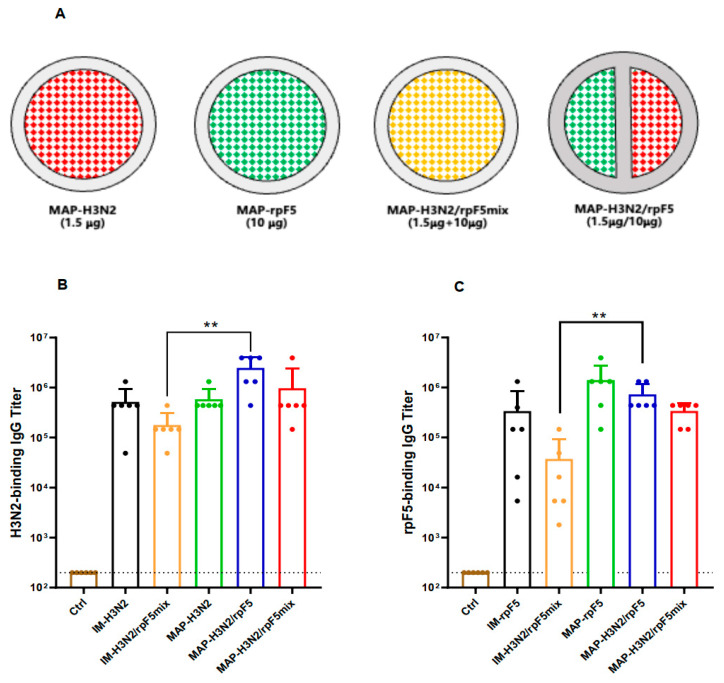
Robust immunogenicity of the MAP-based H3N2/rpF5 combo-preparation. (**A**) Schematic diagrams illustrating the design of D-MAPs carrying the Flu (H3N2) and RSV (rpF5) antigens in either standalone, or premixed, or separately encapsulated forms. Red and blue represent H3N2 and rpF5 antigens, respectively, while yellow the mixture of the two. (**B**,**C**) Groups of BALB/c mice (*n* = 6) were skin-vaccinated twice, on days 0 and 21, with MAP-H3N2 (D-MAPs carrying 1.5 μg/patch A/Darwin split-virus), MAP-rpF5, MAP-H3N2/rpF5mix (D-MAPs laden with premixed A/Darwin split-virus and rpF5), MAP-H3N2/rpF5 (partition-loaded D-MAPs laden with A/Darwin split-virus and rpF5, 1.5 μg:10 μg/patch/dose), or MAP-placebo (Ctrl). Mice i.m. immunized with premixed A/Darwin split-virus and rpF5 (IM-H3N2/rpF5), or standalone A/Darwin split-virus (IM-H3N2), or standalone rpF5 (IM-rpF5) were also included as controls. Serum samples, collected 14 days after the booster immunization, were individually titrated for IgG titers against the A/Darwin Flu virus (**A**) or the rpF5 recombinant protein (**B**) in ELISAs. The data are group mean ± SEM. *p* values were analyzed with two-tailed Mann–Whitney test (**: *p* < 0.01).

**Figure 6 vaccines-13-01213-f006:**
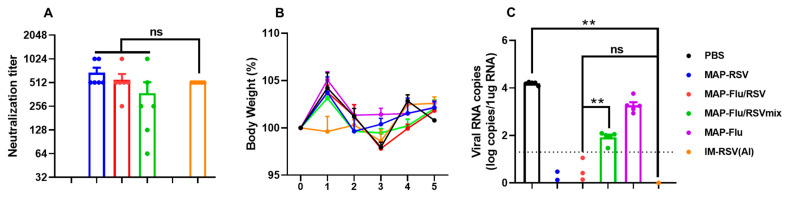
Protection efficacy of the MAP-Flu/RSV combo-vaccine against live RSV challenge in mice. Groups of BALB/c mice (*n* = 6) were given two doses of intradermal vaccination with MAP-Flu/RSV, MAP-Flu/RSVmix, MAP-rpF5, or MAP-Flu, or i.m. immunized with alum-adjuvanted rpF5 (IM-RSV-Al) or PBS alone, followed by assessment of RSV-neutralizing Ab titers in sera (**A**). The vaccinated animals were then i.n. challenged with live RSV, monitored for bodyweight changes until 5 DPI (**B**), and then sacrificed for their lungs for viral load quantitation using qPCR (**C**). Data are group means ± SEM. *p* values were analyzed with two-tailed Mann–Whitney test (**: *p* < 0.01). ns: not significant.

**Figure 7 vaccines-13-01213-f007:**
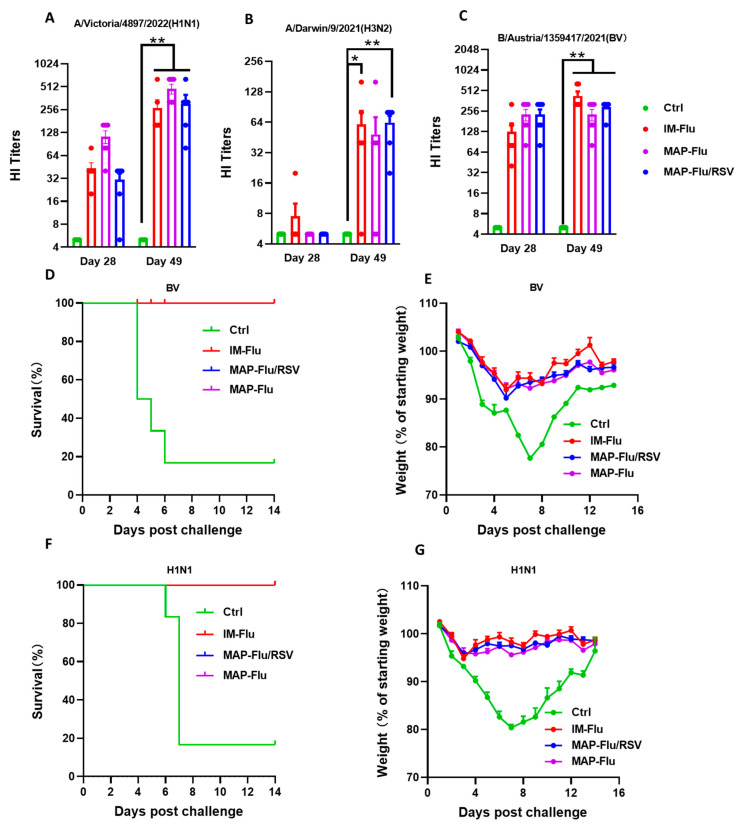
Protection efficacy of the MAP-Flu/RSV combo-vaccine against live Flu viruses in mice. Groups of BALB/c mice (*n* = 6) were either vaccinated, on days 0 and 21, intradermally via MAP skin patch (MAP-Flu/RSV or MAP-Flu) or i.m. with 1.5 μg trivalent Flu vaccine (IM-Flu). A PBS i.m.-injection group was included as additional control (Ctrl). Serum samples, collected on days 28 and 49, were individually titrated for HI titers against Flu virus strains A/Victoria, A/Darwin, and B/Austria (**A**–**C**). The vaccinated mice were i.n. challenged on day 50 with live Flu virus strain A/Victoria, or B/Austria, and monitored for survival rate (**D**,**E**) and bodyweight changes (**F**,**G**) until 14 DPI. For panels except (**D**,**E**), data are group means ± SEM. *p* values were analyzed with two-tailed Mann–Whitney test (*: *p* < 0.05; **: *p* < 0.01).

## Data Availability

Apart from those published here, no new data were created in this study.

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
