# Peer review of "Spatial Segregation Within Dissolving Microneedle Patches Overcomes Antigenic Interference and Enables Potent Bivalent Influenza–RSV Vaccination in Mice"

_vaccines, 2025, doi:10.3390/vaccines13121213_

Round 1
Reviewer 1 Report
Comments and Suggestions for Authors
Specific comments are attached.

The scientific foundation of this manuscript is sound. Authors addressed the challenge of overcoming inter-vaccine interference via the partition-loaded microneedle patch. The findings are compelling, and the authors have demonstrated that the partition-loading strategy successfully overcomes the barrier, which underscores the use of different types of vaccines in a single patch. These findings are significant and exciting for the field.
However, the current manuscript requires a major revision to effectively communicate these findings and sustain reader engagement.
Specifically, the following issues should be addressed:
- In some areas, sentences and paragraphs are excessively long, which obscures the central statements and makes them difficult for the reader to absorb quickly. Please revise the text to be concise and clear. Simplify complex sentence structures. Break long sentences into two or more clearer, tighter statements to improve readability and flow.
- The organization of paragraphs, especially in the Discussion, could be significantly improved. Key arguments and conclusions are sometimes buried or presented in a confusing order. Ensure each paragraph begins with a clear topic sentence that states the main point.
- The text currently reads awkwardly in several places, which distracts from the strength of your data.
Given the technical complexity and high impact of your work, authors may find it useful to get assistance from a native English scientific writer or editing service to ensure the language is professional, clear, and appropriately conveys the significance of your results.
Addressing these points will greatly enhance the impact and accessibility of your valuable research.
Reviewer 2 Report
Comments and Suggestions for Authors
The observations are very interesting, using the microneedle approach to deliver proteins from two viruses (RSV and Flu) for immunization. Interestingly, presenting the same amount of immunogen for both viruses as a mix is not nearly as effective as presenting them separately, though nearby, in the skin. Comments on the manuscript suggest places that the terminology could be simpler and less confusing.
One figure assessing the stabilized form of the prefusion F protein (preF5), reduced and non-reduced, in neighboring lanes was particularly troublesome. The results of this analysis are very different than would be expected. The specifics are thoroughly described in my comments on the manuscript. The problem would likely be resolved by describing the structural modifications in this stabilized F protein. There is no description of how it is stabilized, but there should be.

The English is generally good. Most comments are on English wording for clarity.
Author Response
Comment 1: One figure assessing the stabilized form of the prefusion F protein (preF5), reduced and non-reduced, in neighboring lanes was particularly troublesome. The results of this analysis are very different than would be expected. The specifics are thoroughly described in my comments on the manuscript. The problem would likely be resolved by describing the structural modifications in this stabilized F protein. There is no description of how it is stabilized, but there should be
Response 1: We sincerely appreciate the reviewers' thorough evaluation and constructive feedback, which has significantly strengthened our manuscript. In response to the reviewer’s valuable suggestion, we have expanded the manuscript description of the structural modifications engineered into the stabilized preF5 construct (L250-256). Specifically, the preF5 stabilization strategy incorporates: (i) two strategically placed disulfide bonds (Cys155-Cys290 and Cys149-Cys458) to reinforce key structural domains; (ii) Proline substitutions at position 215 to restrict conformational flexibility; (iii) Cavity-filling mutations (S190F and V207L) to optimize antigenic site stability; (iv) A C-terminal fibritin trimerization domain to ensure native-like trimer formation under physiological conditions. Regarding your reference to "thoroughly described specifics" - we regret that this detailed information does not appear to have been transmitted properly in the review comments.

Round 2
Reviewer 1 Report
Comments and Suggestions for Authors
The authors have addressed all my comments specifically and revised significantly to improve clarity.
At this time, I have only minor comments:
L122, missing “%” after 95
L491, it may be a good idea to have a transitional phrase in the beginning.
For example, ‘despite these compelling results, a few limitations persist/remain in the present study..
Comments on the Quality of English Language
The scientific foundation of this manuscript is sound. Authors addressed the challenge of overcoming inter-vaccine interference via the partition-loaded microneedle patch. The findings are compelling, and the authors have demonstrated that the partition-loading strategy successfully overcomes the barrier, which underscores the use of different types of vaccines in a single patch. These findings are significant and exciting for the field.
However, the current manuscript requires a major revision to effectively communicate these findings and sustain reader engagement.
Specifically, the following issues should be addressed:
- In some areas, sentences and paragraphs are excessively long, which obscures the central statements and makes them difficult for the reader to absorb quickly. Please revise the text to be concise and clear. Simplify complex sentence structures. Break long sentences into two or more clearer, tighter statements to improve readability and flow.
- The organization of paragraphs, especially in the Discussion, could be significantly improved. Key arguments and conclusions are sometimes buried or presented in a confusing order. Ensure each paragraph begins with a clear topic sentence that states the main point.
- The text currently reads awkwardly in several places, which distracts from the strength of your data.
Given the technical complexity and high impact of your work, authors may find it useful to get assistance from a native English scientific writer or editing service to ensure the language is professional, clear, and appropriately conveys the significance of your results.
Addressing these points will greatly enhance the impact and accessibility of your valuable research.
Author Response
Comments: The authors have addressed all my comments specifically and revised significantly to improve clarity. At this time, I have only minor comments: L122, missing “%” after 95; L491, it may be a good idea to have a transitional phrase in the beginning. For example, ‘despite these compelling results, a few limitations persist/remain in the present study..
Response: Thank you very much for your comments. The MS has been further modified as advised.